# Adaptive Softmax Trees for Many-Class Classification

**Rasul Kairgeldin**[1]    **Magzhan Gabidolla**[1]    **Miguel Á. Carreira-Perpiñán**[1]

[1]Department of Computer Science and Engineering , University of California , Merced, CA, USA

## Abstract

NLP tasks such as language models or document classification involve classification problems with thousands of classes. In these situations, it is difficult to get high predictive accuracy and the resulting model can be huge in number of parameters and inference time. A recent, successful approach is the softmax tree (ST): a decision tree having sparse hyperplane splits at the decision nodes (which make hard, not soft, decisions) and small softmax classifiers at the leaves. Inference here is very fast because only a small subset of class probabilities need to be computed, yet the model is quite accurate. However, a significant drawback is that it assumes a complete tree, whose size grows exponentially with depth. We propose a new algorithm to train a ST of arbitrary structure. The tree structure itself is learned optimally by interleaving steps that grow the structure with steps that optimize the parameters of the current structure. This makes it possible to learn STs that can grow much deeper but in an irregular way, adapting to the data distribution. The resulting STs improve considerably the predictive accuracy while reducing the model size and inference time even further, as demonstrated in datasets with thousands of classes. In addition, they are interpretable to some extent.

## 1 INTRODUCTION

Classification problems involving thousands to millions of classes occur naturally in many real-world applications. Examples include predicting the next word in a sentence where the vocabulary size can be in the order of hundreds of thousands, and categorizing products for e-commerce systems where the number of distinct labels can be in the order of millions. Designing fast yet accurate methods for these types of problems remains an active area of research.

A linear softmax model, either standalone or as the last layer in a neural network, is widely used for general classification problems. Its inference time, however, is proportional to the number of classes $K$, as it *needs to evaluate the score for every class no matter the input*, which makes it very slow for large-$K$ classification problems. A natural way to speed it up would be through conditional computation during inference, so that only a small subset of classes needs consideration. Decision trees do this: they *follow a single, instance-dependent root-leaf path during prediction*, and their inference time can potentially be logarithmic on the number of classes. However, traditional axis-aligned trees with constant-label leaves do not produce accurate results for problems with many classes [Choromanska and Langford, 2015].

Recently, Zharmagambetov et al. [2021a] proposed a novel *Softmax Tree (ST)* model that strikes a good balance between linear methods and decision trees: the model takes the form of a (hard) decision tree with sparse oblique (linear) decision nodes and small softmaxes at the leaves. To learn these more complex forms of trees the authors adapt a recent Tree Alternating Optimization (TAO) algorithm [Carreira-Perpiñán and Tavallali, 2018], which can optimize various types of tree-based models but only of a fixed structure and size. Experimentally, STs demonstrate much faster inference than the linear classifier and other baselines, as well as being very accurate for these large classification tasks. However, *a significant drawback is that it assumes a complete tree structure, whose size grows exponentially with depth, and this limits their power in both accuracy and inference time*. We discuss this in detail in section 3, where we show that *the key to achieve fast inference time is to decrease the size of the leaf softmaxes by increasing the depth of the leaf path*. Thus, we propose a new model, *Adaptive Softmax Trees (ASTs)*, where we learn jointly the structure and parameters of the tree, by interleaving steps that grow the structure optimally with steps that optimize the parameters of the current structure. This

makes it possible to learn ASTs that can grow much deeper but in an irregular way, adapting to the data distribution. As we show experimentally, the resulting ASTs improve considerably the predictive accuracy while reducing the number of parameters and inference time even further.

We now review related work (section 2), discuss the difficulty of searching over tree structures (section 3), and describe the original softmax tree (ST) model and TAO-based optimization (section 4) and our proposed adaptive softmax trees (AST) (section 5). Then (section 6) we experimentally show the superiority of ASTs over STs and other baselines for several multi-class classification problems with a large number of classes and for language modeling.

## 2 RELATED WORK

### 2.1 SOFTMAX APPROXIMATION

While a softmax linear classifier defines a convex problem with the cross-entropy, it has long been recognized that training it with many classes is a huge computational bottleneck, so that one-vs-all can often be the only affordable option, in part due to its inherent parallelism [Deng et al., 2010]. Indeed, even the widely used, extremely efficient LIBLINEAR [Fan et al., 2008] implements one-vs-all but not the cross-entropy softmax. And, once trained, inference time in a large softmax is also very large—for example, in a language model having a large vocabulary. Hence, much work has been devoted to approximating the softmax classifier. The Hierarchical Softmax (HSM) [Goodman, 2001] addresses this by using a predetermined tree structure with linear decision nodes and fixed leaf labels (corresponding to the words in vocabulary) to speed up the training of language models. Originally developed for a two-level tree, it has been extended to deeper architectures [Morin and Bengio, 2005, Mnih and Hinton, 2009]. The structure of the tree can be random, or based on word similarities [Brown, 1992, Le et al., 2011, Mikolov et al., 2013b], or on word frequencies [Mikolov et al., 2013a, Le et al., 2013], or based on speed-optimal dynamic programming [Zweig and Makarychev, 2013], and optimized for GPUs [Grave et al., 2017]. Training HSM-based language models is efficient (usually logarithmic in vocabulary size), but it leads to no speedup at inference time: during prediction, although some pruning is possible, an input instance is propagated to nearly all the leaves. Apart from HSMs, other methods of softmax approximation are possible, such as singular value decomposition [Shim et al., 2017] and model compression techniques such as pruning or quantization [Deng et al., 2020].

### 2.2 DECISION TREE METHODS

Decision trees enjoy fast prediction and interpretability, but traditional methods such as CART [Breiman et al., 1984] have low accuracy for problems with many classes [Choromanska and Langford, 2015]. This is due to two reasons: firstly, a suboptimal training based on greedy recursive partitioning, where the tree parameters are fixed using a local heuristic as one grows the tree (so the result is not optimal in any sense); and secondly, a limited modeling ability because of using trees with axis-aligned splits (which are poorly suited for high-dimensional data) and constant-label leaves. The trees can be made more complex by allowing for oblique (hyperplane) splits [Breiman et al., 1984] and small linear classifiers at the leaves [Daumé III et al., 2017]. However, this leads to a more difficult optimization problem for which various heuristics have been proposed within the many-class setting [Jernite et al., 2017, Daumé III et al., 2017]. Zharmagambetov et al. [2021a] use Tree Alternating Optimization to learn these models, but it is limited to trees of fixed structure. Decision trees are usually ensembled to boost accuracy but traditional implementations are not suitable for problems with many classes. Si et al. [2017] adapt gradient boosting trees to output $\ell_0$-regularized sparse prediction and apply this to many-class problems. Besides tree-based techniques, other methods exist such as sampling [Joshi et al., 2017] and hashing [Medini et al., 2019].

Instead, we build on the line of work initiated by the Tree Alternating Optimization (TAO) algorithm [Carreira-Perpiñán and Tavallali, 2018] (described in section 4) and follow-up works. TAO is able to optimize a very general objective function over the parameters of a fixed-structure tree by repeatedly optimizing each node given the rest are fixed, and it achieves trees that are smaller but more accurate than traditional ones [Zharmagambetov et al., 2021b]. It makes it possible to train new types of trees, such as having sparse oblique splits [Carreira-Perpiñán and Tavallali, 2018], bivariate splits [Kairgeldin and Carreira-Perpiñán, 2024], or having neural nets in the leaves [Zharmagambetov and Carreira-Perpiñán, 2021]. The ability to use different loss functions makes it possible to learn trees optimally for tasks where they were never or rarely used before, such as clustering [Gabidolla and Carreira-Perpiñán, 2022b], dimensionality reduction [Zharmagambetov and Carreira-Perpiñán, 2022a], semi-supervised learning [Zharmagambetov and Carreira-Perpiñán, 2022b], imbalanced classification [Gabidolla et al., 2024], or for probing the meaning of individual neurons in neural nets via model distillation [Hada et al., 2023], among others. Finally, TAO also makes it possible to ensemble trees into forests for classification or regression using bagging [Carreira-Perpiñán and Zharmagambetov, 2020, Zharmagambetov and Carreira-Perpiñán, 2020], boosting [Gabidolla et al., 2022, Gabidolla and Carreira-Perpiñán, 2022a] or even a joint global optimization over all trees [Carreira-Perpiñán et al., 2023], which results in forests that are smaller but more accurate than traditional, axis-aligned ones, such as XGBoost [Chen and Guestrin, 2016] or LightGBM [Ke et al., 2017].

## 2.3 CONDITIONAL COMPUTATION

There is growing interest in having neural networks use only a small portion of their computational graph to enable fast prediction. Although several works [Shazeer et al., 2017, Hazimeh et al., 2020, Veit and Belongie, 2018] have shown promising results in terms of runtime and accuracy tradeoff, the non-differentiability of the conditional computation makes it difficult to apply gradient-based optimization [Hazimeh et al., 2021]. One way to achieve this is to train a continuous model, such as a soft tree, and harden its decisions a posteriori, but this leads to degradation in accuracy, as observed by Zharmagambetov et al. [2021a]. In our adaptive softmax trees, conditional computation is built in by design during training and inference.

## 2.4 GROWING NEURAL NETS AND NEURAL ARCHITECTURE SEARCH

The idea of growing the neural architecture by adding more neurons during training has a long history [Fahlman and Lebiere, 1990, Gallant, 1993, Fritzke, 1994, Bruske and Sommer, 1995, Evci et al., 2022]. A related, recently very active area that aims to learn an optimal neural net structure is Neural Architecture Search (reviewed by Elsken et al. [2019], Ren et al. [2021]). A major issue in learning/growing neural architectures is the vast number of choices: layerwise or depthwise growth, how to connect neurons, etc. With trees the search space is more directed: either one expands leaves or prunes nodes. Tanno et al. [2019] adaptively grow and train neural trees using backpropagation, but the potential gains in inference speed are limited: firstly, their trees are soft, so an input instance has to follow all root-leaf paths (each with a positive probability), which is proportional to the number of leaves (typically exponential on the depth); and secondly, their trees are very small, having just a few leaves, which forces each node to use a relatively large neural net, so even if we follow a single path (as a fast approximation) it will still be computationally costly. In contrast, a deeper tree, having many, deep, lighter leaves, is faster at inference, and can still have high accuracy, as we show here with our adaptive softmax trees.

## 3 OPTIMIZING TREES OVER PARAMETERS AND STRUCTURES: DEEP PATHS, THIN SOFTMAXES

Learning a tree-based model has two important difficulties. One is that the space of tree structures is huge: with $n$ nodes (in total), there are $\frac{1}{n+1}\binom{2n}{n}$ ordered trees [Knuth, 1997], which already exceeds one million for $n = 14$. The other is that a (hard) tree defines a non-differentiable, highly non-convex optimization problem.

The traditional, widely used approach for learning axis-aligned trees is based on *greedy top-down induction* [Breiman et al., 1984, Quinlan, 1993]: starting from the root node, splits are recursively fixed (to optimize a local purity criterion) until the tree is fully grown. This is usually followed by a form of pruning to reduce overfitting. While suboptimal, this two-step process does effect a form of local search over tree structures and can produce adequate results with simple axis-aligned constant-leaf trees, but it works poorly with more complex trees, e.g. with oblique or neural nodes.

The *Tree Alternating Optimization (TAO)* algorithm [Carreira-Perpiñán and Tavallali, 2018], reviewed in section 4 for Softmax Trees, works by optimizing the parameters of each node in alternation, for a tree of a given structure. It does a much better job at optimizing a complex tree, as it can monotonically decrease a loss function, regularization term, and node models of general form. It also does a restricted form of structure search: an $\ell_1$ penalty sparsifies the node weight vectors, which can make nodes redundant and thus pruned, resulting in a learned structure that is a subtree of the initial tree. But, beyond that, TAO does not search over tree structures, and in particular, it cannot learn a bigger tree than the initial one.

The original Softmax Tree [Zharmagambetov et al., 2021a], consisting of a tree with oblique (hard) splits and softmax leaves, relied directly on TAO to optimize the cross-entropy. As an initial tree, it used a complete tree of depth $\Delta$ and $2^{\Delta}$ softmaxes each having $k$ classes. By tuning these two hyperparameters $\Delta$ and $k$, it achieved good results on large, many-class datasets. But it has a major limitation: *the number of nodes grows exponentially with the depth, which is thus computationally limited in memory and time (to $\Delta \approx 14$ in that paper), which in turn forces the softmaxes to use many classes ($k$ up to 800 in that paper)*. If the tree was deeper, the softmaxes could be smaller, accelerating the inference. Crucially, depending on the data distribution, the tree may need to be quite deep in some parts and shallow in others, i.e., an unbalanced structure. If we could guess the right structure, we could have TAO use that from the beginning, but guessing it is far from simple. Using, say, a structure from a CART tree does not work at all. This calls for searching over structures properly as proposed in our *Adaptive Softmax Trees (ASTs)*, described in section 5. And, as it turns out, we find in our experiments that ASTs achieve higher test accuracy than using a complete ST of the same depth (which is far more costly).

At the heart of the improvement of ASTs is the interplay between tree depth $\Delta$ and leaf softmax width $k$. Let $D \in \mathbb{N}$ be the feature dimensionality. In a complete ST, the inference time is $\mathcal{O}(D(\Delta + k))$ (actually less if the weight vectors and softmaxes are sparse and some tree paths are shallower than $\Delta$), and typically $\Delta \ll k$. This already improves greatly over a single softmax, $\mathcal{O}(DK)$, if $\Delta + k \ll K$. In

ASTs, an irregular tree structure makes it possible to reduce $k$ further by increasing $\Delta$ selectively for each branch. Besides, our ASTs learn the number of classes $k_j$ for each leaf $j$ automatically, so that some leaves specialize on a few select classes while others handle more, which affords more speedups. The inference time is then $\mathcal{O}(D(\Delta_j + k_j))$ for each leaf $j$, and usually larger $\Delta_j$ are associated with smaller $k_j$.

## 4 SOFTMAX TREES (STs) AND TREE ALTERNATING OPTIMIZATION (TAO)

We now describe the Softmax Tree (ST) model and the extension of TAO to train them over a fixed tree structure [Zharmagambetov et al., 2021a]. Let $\{(\mathbf{x}_n, y_n)\}_{n=1}^N \subset \mathbb{R}^D \times \{1, \ldots, K\}$ be our training set of size $N$ of $D$-dimensional input features and $K$ classes. Write the *Softmax Tree* as $\boldsymbol{\tau}(\mathbf{x}; \boldsymbol{\Theta})$, a rooted binary tree with a set of decision (internal) nodes $\mathcal{N}_{\text{dec}}$ and a set of leaf nodes $\mathcal{N}_{\text{leaf}}$. Each decision node $i \in \mathcal{N}_{\text{dec}}$ has a decision function $g_i(\mathbf{x}; \boldsymbol{\theta}_i) \colon \mathbb{R}^D \to \{\texttt{left}_i, \texttt{right}_i\} \subset \{\mathcal{N}_{\text{dec}} \cup \mathcal{N}_{\text{leaf}}\}$ that sends an instance $\mathbf{x}$ to its left or to its right child. We use oblique (linear) decision nodes: "if $\mathbf{w}_i^T \mathbf{x} + w_{i0} \geq 0$ then $g_i(\mathbf{x}) = \texttt{right}_i$, otherwise $g_i(\mathbf{x}) = \texttt{left}_i$" where the learnable parameters are $\boldsymbol{\theta}_i = \{\mathbf{w}_i, w_{i0}\}$. Note how the decision function makes hard decisions, unlike in soft trees, where an instance $\mathbf{x}$ is propagated to both children with a positive probability. Each leaf $j \in \mathcal{N}_{\text{leaf}}$ contains a predictive function $\mathbf{f}_j(\mathbf{x}; \boldsymbol{\theta}_j) \colon \mathbb{R}^D \to \mathbb{S}^K$ that produces the actual output of the tree $\boldsymbol{\tau}(\mathbf{x}; \boldsymbol{\Theta})$ for an instance $\mathbf{x}$, where $\mathbb{S}^K = \{\mathbf{x} \in [0, 1]^K \colon \mathbf{1}^T \mathbf{x} = 1\}$. In Softmax Trees, $\mathbf{f}_j(\mathbf{x}; \boldsymbol{\theta}_j)$ takes the form of a small softmax linear classifier: $\mathbf{f}_j(\mathbf{x}; \boldsymbol{\theta}_j) = \sigma(\mathbf{W}_j \mathbf{x} + \mathbf{w}_{j0})$ where $\boldsymbol{\theta}_j = \{\mathbf{W}_j \in \mathbb{R}^{k \times D}, \mathbf{w}_{j0} \in \mathbb{R}^k\}$ are the learnable parameters, and $\sigma(\cdot)$ is the softmax function. The leaf predictor function $\mathbf{f}_j(\mathbf{x}; \boldsymbol{\theta}_j)$ can output only $k$ nonzero probabilities, with $k \leq K$, for a set of $k$ classes (this set is learned); for all the other $K - k$ classes $\mathbf{f}_j(\mathbf{x}; \boldsymbol{\theta}_j)$ assigns exactly zero probability. For problems with a large number of classes we want $k \ll K$ to allow for fast inference. The predictive function of the whole Softmax Tree $\boldsymbol{\tau}(\mathbf{x}; \boldsymbol{\Theta})$ then works by routing an instance $\mathbf{x}$ to exactly one leaf through a root-leaf path of (oblique) decision nodes and applying that leaf's small softmax predictor function. Overall, a ST can be seen as a hierarchical collection of local softmax classifiers each operating on a small subset of classes.

Now we describe how the TAO algorithm applies in learning a ST. TAO is a general method for optimizing a given objective function over a given decision tree model. Unlike CART-type methods, TAO works similarly to how one would optimize a (say) neural network: by taking an initial tree structure (cf. network architecture) and parameters (cf. network weights) it performs alternating optimization over the nodes (cf. gradient descent in a neural net) to mono-

tonically decrease the objective function. Unlike with neural nets and soft decision trees, gradient-based optimization is not applicable because hard decision trees are non-differentiable functions. Given a Softmax Tree $\boldsymbol{\tau}(\mathbf{x}; \boldsymbol{\Theta})$ of fixed structure (e.g. a complete tree of depth $\Delta$) and initial parameters (e.g. random), the goal of TAO is to minimize the following objective:

$$E(\boldsymbol{\Theta}) = \sum_{n=1}^N L(\mathbf{y}_n, \boldsymbol{\tau}(\mathbf{x}_n)) +$$
$$\lambda \sum_{i \in \mathcal{N}_{\text{dec}}} \|\mathbf{w}_i\|_1 + \mu \sum_{j \in \mathcal{N}_{\text{leaf}}} \|\mathbf{W}_j\|_1 \quad (1)$$

where $L(\cdot, \cdot)$ is the cross-entropy loss, $\boldsymbol{\Theta} = \{\mathbf{w}_i, w_{i0}\}_{i \in \mathcal{N}_{\text{dec}}} \cup \{\mathbf{W}_j, \mathbf{w}_{j0}\}_{j \in \mathcal{N}_{\text{leaf}}}$ are the set of all learnable model parameters, and there is an $\ell_1$ penalty over the weight vectors to promote sparsity via hyperparameters $\lambda, \mu \geq 0$. In general, we use the same regularization value for both decision nodes and leaves $\lambda = \mu$, but in some experiments we explore the effect of the leaf sparsity $\mu$.

The TAO algorithm is based on two theorems. First, the *separability condition* states that eq. (1) separates over a set of non-descendant nodes, e.g. all the nodes at a given depth. This is a consequence of the tree making hard decisions. All such non-descendant nodes can be optimized independently and in parallel. Second, the *reduced problem over a node* states that optimizing the top-level problem of eq. (1) over the parameters of a given node $i \in \{\mathcal{N}_{\text{dec}} \cup \mathcal{N}_{\text{leaf}}\}$ reduces to a simpler, well-defined problem involving only the training instances that currently reach that node $i$ (the *reduced set* $\mathcal{R}_i \subset \{1, \ldots, N\}$). The exact form of the reduced problem differs for leaves and for decision nodes:

- For a decision node $i \in \mathcal{N}_{\text{dec}}$, the top-level problem of eq. (1) reduces to a *weighted 0/1 loss binary classification problem*:

$$E_i(\mathbf{w}_i, w_{i0}) = \sum_{n \in \mathcal{R}_i} c_n \overline{L}(\overline{y}_n, g_i(\mathbf{x}_n)) + \lambda \|\mathbf{w}_i\|_1 \quad (2)$$

where $\overline{L}(\cdot, \cdot)$ is the 0/1 loss, $\overline{y}_n \in \{\texttt{left}_i, \texttt{right}_i\}$ is a pseudolabel indicating the "best" child (i.e., the child that gives the lower value of the loss down its subtree) for the instance $\mathbf{x}_n$, and $c_n \geq 0$ is the loss difference between the "other" child and the "best" child for the instance $\mathbf{x}_n$. This problem over an oblique node is in general NP-hard, but it can be approximated well with a surrogate loss such as the cross-entropy (i.e., solving a logistic regression). We can ensure a monotonic decrease of the top-level objective (1) by accepting the update only if it improves (2) (in practice we find this unnecessary).

- For leaf node $j \in \mathcal{N}_{\text{leaf}}$, the top-level problem of eq. (1) reduces to a form involving the original loss but only over the parameters of the leaf predictor function $\mathbf{f}_j(\cdot)$ and its

reduced set $\mathcal{R}_j$:

$$E_j(\mathbf{W}_j, \mathbf{w}_{j0}) = \sum_{n \in \mathcal{R}_j} L(\mathbf{y}_n, \mathbf{f}_j(\mathbf{x}_n)) + \mu \left\| \mathbf{W}_j \right\|_1 \quad (3)$$

where $L(\cdot, \cdot)$ is the same cross-entropy loss of eq. (1). Exactly solving this problem would require enumerating all $\binom{K}{k}$ class subsets, but we can approximate this well by picking the top $k$ majority classes in the reduced set $\mathcal{R}_j$ and training a $k$-class softmax classifier $\mathbf{f}_j(\cdot)$ on them. We solve the resulting $\ell_1$-regularized convex problem using SAG [Schmidt et al., 2017].

While these theorems do not prescribe the order in which the nodes should be optimized, Zharmagambetov et al. [2021a] follow a reverse breadth-first search order: all the nodes at a given depth are optimized in parallel, starting from the deepest ones until the root. Each optimization subproblem involves solving either an $\ell_1$-regularized logistic regression or an $\ell_1$-regularized $k$-class softmax classifier. As an initial tree, a complete tree of a given depth $\Delta$ is used with initial parameters set either randomly or based on a $k$-means clustering assignment of training points to the leaves. The hyperparameters of the model are the depth $\Delta$ of the tree and the number of classes $k$ in each of the leaf softmaxes. Fig. 1 (left) outlines the pseudocode of TAO for Softmax Trees. By ensuring that the (approximate) solution of the reduced problem of a decision node improves upon the previous node parameter values, TAO is guaranteed to decrease the objective function (1) monotonically.

Finally, node pruning occurs automatically because the $\ell_1$ penalty can drive a node's entire weight vector to zero. This makes the node redundant (it sends all instances to the same child) and it can be removed at the end. Thus, the final ST is a subset of the initial (complete) ST.

## 5 ADAPTIVE SOFTMAX TREES (ASTs)

In the previous section, TAO was used on a complete tree of depth $\Delta$. Now, we improve this to explore structures. The basic idea is to use two types of steps. One is a *regular* TAO optimization of a ST of fixed structure (not necessarily complete); this guarantees improvement of the objective defined *globally* over this ST. The other is an *expansion step* on a current leaf, which tries to replace it with a shallow ST (having narrower softmaxes at its leaves). This *local move* can improve the loss function but at the cost of additional decision nodes; we actually expand the leaf if overall we improve, else we do not expand it, and try another leaf. We interleave regular and expansion steps until convergence. Let us see this in more detail.

We first train a shallow (e.g. depth $\Delta = 2$) complete Softmax Tree $\boldsymbol{\tau}(\cdot; \boldsymbol{\Theta})$ with relatively large $k_0$-class softmaxes in the leaves. The number of classes $k_0$ is set such that the total number of predictable classes by the model is at least

the total number of classes $K$ in the dataset: $k_0 2^\Delta \geq K$. We then attempt to replace each leaf $j \in \mathcal{N}_{\text{leaf}}$ softmax predictor function $\mathbf{f}_j(\cdot; \boldsymbol{\theta}_j)$ by yet another shallow Softmax Tree $\hat{\boldsymbol{\tau}}_j(\cdot; \hat{\boldsymbol{\Theta}}_j)$ of depth $\hat{\Delta} = 1$ or 2, whose leaves contain smaller $\hat{k}_j$-class softmaxes, $\hat{k}_j < k_0$. To control by how much these large softmaxes are reduced we use the following simple heuristic: $\hat{k}_j = \alpha \, k_0$, where $\alpha \in (0, 1)$ is the *softmax contraction coefficient hyperparameter*. We obtain this small tree $\hat{\boldsymbol{\tau}}_j(\cdot; \hat{\boldsymbol{\Theta}}_j)$ by fitting it using the TAO algorithm on the training instances that reach the leaf $j$, i.e., on the reduced set $\mathcal{R}_j$. This step can be considered as a recursive application of the Softmax Tree method with the goal of replacing large, flat softmaxes with faster "softmax subtrees". But instead of directly substituting the leaf softmax $\mathbf{f}_j(\cdot; \boldsymbol{\theta}_j)$ with the tree $\hat{\boldsymbol{\tau}}_j(\cdot; \hat{\boldsymbol{\Theta}}_j)$, we first ensure that the accuracy of $\hat{\boldsymbol{\tau}}_j(\cdot; \hat{\boldsymbol{\Theta}}_j)$ is at least as good as the original softmax $\mathbf{f}_j(\cdot; \boldsymbol{\theta}_j)$ or within a reasonable *tolerance ratio hyperparameter* $\rho > 1$. If this is not the case, the leaf predictor function $\mathbf{f}_j(\cdot; \boldsymbol{\theta}_j)$ remains unchanged. Otherwise, the substitution happens, and this results in the structure change of the original tree model $\boldsymbol{\tau}(\cdot; \boldsymbol{\Theta})$ where it is expanded through the leaf $j$ (the *expansion step*). In this way, after attempting to expand all the leaves $j \in \mathcal{N}_{\text{leaf}}$, and assuming some or all of them are expanded, we obtain a deeper, irregular Softmax Tree $\boldsymbol{\tau}_{\text{exp}}(\cdot; \boldsymbol{\Theta}_{\text{exp}})$ with smaller leaf softmaxes which has comparable or better training accuracy and faster inference. Now, importantly, we retrain the whole model $\boldsymbol{\tau}_{\text{exp}}(\cdot; \boldsymbol{\Theta}_{\text{exp}})$ globally using TAO (the *regular step*), which will further improve the model accuracy and possibly sparsify nodes. We repeat these local expansion and global optimization steps until the model converges or some predetermined stopping criterion is reached. Note that if a given leaf $j$ could not grow at one expansion step, it can still grow in the next iteration because of the in-between optimization step which can change the parameters of the whole model. Fig. 1 outlines the proposed adaptive learning algorithm.

Note that the expansion move allows us to compare the objective function before and after the expansion in order to decide whether or not we should pursue a new architecture. This is possible by expanding a leaf subtree and optimizing it separately, which in turn is possible because of the separability condition that trees satisfy: the objective function separates additively over the leaf subtrees, because each leaf subtree operates only on its own parameters, its own region of the input space and its own reduced set (training instances reaching the leaf). Thus, the contribution to the overall (tree-wide) objective function of optimizing over an expanded leaf subtree is a separable term. Comparing the loss on the leaf reduced set before (softmax) and after (softmax subtree optimized only on that reduced set), together with the regularization term on the parameters, gives the exact improvement of the overall objective function. This makes it possible to decide locally whether to accept the expansion or not. The subsequent, global optimization with TAO of the expanded tree may, of course, undo some of

```
input training set {x_n, y_n}_{n=1}^N,
   Softmax Tree τ(·; Θ) of depth Δ.
repeat
   update reduced sets R_i for all nodes i;
   for d = Δ downto 0 do
      for i ∈ nodes at depth d
         if i is a leaf
            fit a k_i-class linear softmax f_i(·; θ_i)
            on the top-k_i majority class points
            in R_i to optimize eq. (3)
         else
            fit a weighted 0/1 loss binary
            classifier g(·; θ_i) to optimize eq. (2)
         end if
      end for
   end for
until stopping criterion
return trained τ(·; Θ)
```

```
input training set {x_n, y_n}_{n=1}^N, initial depth Δ_0,
   softmax contraction coefficient α ∈ (0, 1),
   tolerance ratio for node expansion ρ > 1.
k_0 ← αK
initialize τ(·; Θ) of depth Δ_0 and k_0-class leaves;
fit τ(·; Θ) using TAO;
repeat
   update reduced sets R_j for all j ∈ N_leaf;
   for j ∈ N_leaf
      initialize ST τ̂_j(·; Θ̂_j) of depth Δ = 1 or 2
      and with (αk_j)-class softmax leaves;
      fit τ̂_j(·; Θ̂_j) using TAO on {x_n, y_n}_{n∈R_j};
      if  loss(τ̂_j(·;Θ̂_j)) / loss(f_j(·;θ_j)) < ρ  then accept the expansion of leaf j
   end for
   update the tree τ(·; Θ) and reoptimize with TAO;
until no changes to the tree structure
return adaptively grown τ(·; Θ)
```

Figure 1: *Left*: pseudocode of TAO for learning Softmax Trees. *Right*: pseudocode of the proposed adaptive growth method for ASTs; this uses TAO (left part) as a subroutine.

the expansions, as well as update all the parameters and reduced sets. Also, the local expansion moves are fast thanks to using the existing weight matrix to warm-start the optimization in the leaves.

This algorithm can be motivated as performing a search through the vast space of different tree structures and parameters. Each leaf-wise local expansion step tries to improve the tree structure, and the subsequent optimization step of the whole current tree tries to refine its parameters. This process leads to a better structure, often highly irregular and far from complete, and better parameters than the one produced by TAO on a random or heuristic complete tree initialization. The hyperparameters $\alpha$ and $\rho$ are designed to control how fine the search over structures is: the smaller $\alpha$ the faster the softmaxes contract (so shallower trees), and the smaller $\rho$ the more accurate the expanded subtree must be to be accepted. They also help to control overfitting.

### 5.1 COMPUTATIONAL COMPLEXITY OF ASTs

**Training** It is difficult to estimate the training time precisely because of the changing tree structure and softmax sizes. A coarse upper bound results from taking the largest structure and softmax size $k_{max}$ that occur during training. If we assume that fitting softmax classifiers is linearly proportional to the training set size, then *sequential* optimization of all the leaves is upper-bounded by fitting a single $k_{max}$-class softmax for the whole training set. But after several expansion steps, the softmax sizes are usually much smaller. Regarding the oblique decision nodes, opti-

mizing *sequentially* all of them at a given depth is asymptotically equivalent to fitting a single logistic regression on the whole training set. However, from TAO's separability condition, optimizing all the leaves and all the decision nodes at the same depth can be done *in parallel*, which can bring huge speedups.

**Inference** For the original Softmax Tree (assumed complete), the inference time is $\mathcal{O}(D(\Delta + k))$. Compared to a single flat softmax on all $K$ classes, the speed-up is dramatic: $\mathcal{O}(\frac{K}{\Delta+k}) \approx \mathcal{O}(\frac{K}{k})$ if $k \approx \Delta+k \ll K$. For our AST, the inference time for a leaf $j$ is $\mathcal{O}(D(\Delta_j + k_j))$. The improvement is that this results in quite smaller values of $k_j$ at the expense of slightly large values of $\Delta_j$ (thin softmaxes in deep leaves).

## 6 EXPERIMENTS

Our experimental results consistently demonstrate the benefit of our proposed adaptive learning method in learning better Softmax Trees in terms of accuracy, inference time, and model size for several benchmarks in classification tasks with a large number of classes and in language modeling. After describing our setup, we first show a detailed comparison of the proposed adaptive growth method against the previous fixed tree approach. We then report benchmark results for document classification and language modeling tasks. Finally, we analyze the produced tree structure and attempt to interpret the model by visualizing it. In this section, "AST" refers to our proposed adaptive learning method, and "ST" refers to the previous fixed tree ap-

proach.

## 6.1 SETUP

Unless otherwise stated, we use the following fixed values for these hyperparameters: the initial tree depth $\Delta_0 = 2$ and the depth of expanding subtrees $\hat{\Delta} = 1$. For all other hyperparameters (the sparsity of decision nodes and leaves $\lambda = \mu$, tolerance ratio for node expansion $\rho$ and softmax contraction coefficient $\alpha$) we set them in accordance with cross-validation on a holdout set. All other implementation details including hyperparameter tuning are provided in appendix A.2.

We compare our results with other baselines specifically developed for problems with a large number of classes. These include RecallTree [Daumé III et al., 2017], $(\pi, \kappa)$-DS [Joshi et al., 2017] and MACH [Medini et al., 2019]. We use available open-source implementations of the above methods or cite their results, where applicable. As noted in the related work section, training a softmax classifier by optimizing the cross-entropy is very time-consuming, so in some of our comparisons we use one-vs-all classifiers because training a linear softmax classifier was infeasible even in our 256GB server. For the linear one-versus-all classifier we use `scikit-learn`'s implementation [Pedregosa et al., 2011]. In contrast, our ASTs can be trained with fewer resources and are much faster at inference. Compared to the original ST, ASTs can grow much deeper with very narrow softmax layers, even using just one class ($k = 1$) in some leaves. We report misclassification errors on train and test sets, average inference time per sample on test set, and tree parameters (tree depth $\Delta$, average leaf softmax sizes $\bar{k}$, and the number of leaves). We time the inference of each sample on a single CPU and average it over the whole test set. All experiments are conducted on the machine Intel Xeon CPU E5-2699 v3 @ 2.30GHz, 256 GB RAM.

## 6.2 THE BENEFIT OF ADAPTIVE GROWTH

We first perform a detailed comparison between the models produced by our adaptive growth method and the previous fixed tree approach. We use the following datasets with a large number of classes: WIKI-Small subs., ALOI, LSHTC1. The details about them are in appendix A.1.

For these sets of controlled experiments, we keep node and leaf sparsity parameters $\lambda, \mu$ equal for both ASTs and STs. As stated in previous sections, the AST approach expands leaves unevenly, which produces softmaxes with different number of classes $k$. To ensure that the comparison between resulting models is fair and comprehensive, we train STs with the biggest $k$ from an AST and cross-validated depth $\Delta$. For WIKI-small we provide a pairwise comparison of multiple STs and ASTs of similar $k$ in Table 1. For

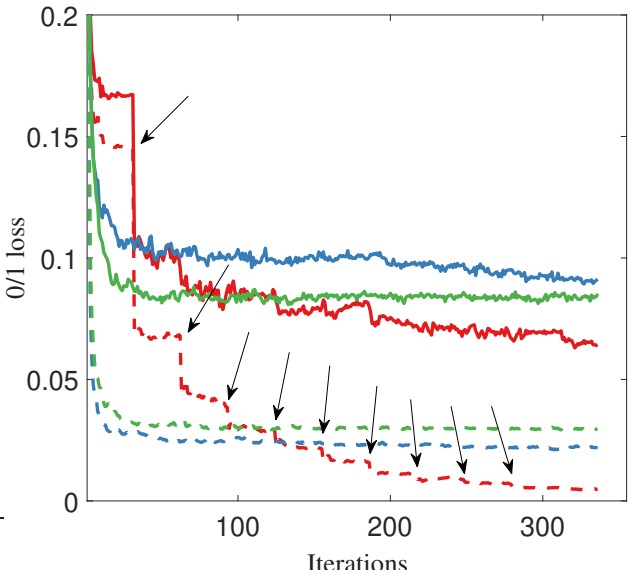

Figure 2: 0/1 loss of the final AST model for training (dashed line) and test (solid line), compared with the complete Softmax Tree. The arrows point to where expansions of the AST happened. The line colors indicate the performance of the ST (blue), ST(AST) (green) and AST (red). This shows that the adaptive growth gradually enhances the performance of the model on both training and test tests (red solid and dashed lines). On the other hand, a ST initialized randomly (blue line) or on the final structure of AST (green line) is unable to improve after a certain number of iterations.

example, the softmax size of $ST^*(k = 13)$ and a maximum softmax size of $AST^*(\alpha = 0.39, \rho = 1.2)$ are equal. Then, we use the structure of the final tree from the AST to initialize an ST (referred as "ST(from AST)"). We keep $k$ of leaf softmaxes but reinitialize randomly the weights of linear classifiers in decision nodes and leaves.

Table 1 shows that ASTs considerably outperform STs in test error (up to 5% on WIKI-Small). In many cases, the performance of ST is improved as we lower the depth but lowering it too much leads to an increase in test error. Note that the depth of STs initialized from the corresponding AST differs because of the post-pruning. Importantly, ASTs have much faster inference (up to 15 times) and lower FLOPs. Fig. 2 contains an additional experiment showing the improved accuracy of ASTs over STs as a function of training iterations. These sets of experiments confirm that the progressive growth of a tree results in a better local optimum and justifies our proposed approach.

## 6.3 TEXT CLASSIFICATION

We compare our method with other baselines (including ST) on document categorization benchmark WIKI–Small

Table 1: AST vs ST. We report: train/test errors; depth $\Delta$, number of leaves $L$, average leaf softmax size $\bar{k}$ of the tree; and average inference time and FLOPs per test instance. For ST we specify its leaf softmax size $k$, for AST the softmax contraction coefficient $\alpha$ and tolerance ratio of expansion $\rho$. ASTs are trained with $\mu = 0.01$ or (if marked with $*$) $\mu = 0.1$.

| | Method | $E_{\text{train}}\%$ | $E_{\text{test}}\%$ | $\Delta$ | $L$ | $\bar{k}$ | inf.(μs) | FLOPs |
|---|---|---|---|---|---|---|---|---|
| Letter | Softmax | 22.30 | 23.20 | – | – | – | 53 | 416 |
| | ST($k=7$) | 0.52 | 8.33 | 7 | 128 | 5.27 | 142 | 197 |
| | ST($k=5$) | 0.36 | 8.75 | 8 | 256 | 3.53 | 98 | 214 |
| | ST(from AST) | 2.94 | 8.84 | 11 | 373 | 1.77 | 86 | 177 |
| | **AST**($\alpha$=0.85,$\rho$=1.2) | 0.30 | 7.03 | 12 | 153 | 2.13 | 43 | 162 |
| | **AST**($\alpha$=0.75,$\rho$=1.2) | 2.05 | **6.35** | 15 | 384 | 1.01 | **9** | **151** |
| ALOI | Softmax | 10.90 | 13.0 | – | – | – | 411 | 128000 |
| | ST$^*$($k=90$) | 2.01 | 12.3 | 7 | 126 | 64.9 | 24 | 1493 |
| | ST($k=75$) | 3.89 | 12.0 | 6 | 64 | 74.9 | 29 | 1871 |
| | ST(from AST) | 2.37 | 12.8 | 8 | 177 | 38.4 | 18 | 1102 |
| | **AST**$^*$($\alpha$=0.75,$\rho$=1.01) | 1.49 | **9.9** | 10 | 326 | 23.8 | **15** | **1016** |
| LSHTC1 | Softmax | 54.30 | 61.4 | – | – | – | 10680 | 423722 |
| | ST($k=70$) | 14.20 | 62.7 | 7 | 128 | 70.0 | 65 | 12279 |
| | ST($k=50$) | 6.15 | 61.2 | 8 | 256 | 49.4 | 55 | 9218 |
| | ST(from AST) | 9.36 | 68.7 | 9 | 511 | 49.7 | 62 | 9388 |
| | **AST**$^*$($\alpha$=0.9,$\rho$=1.2) | 16.10 | **60.8** | 10 | 1006 | 11.5 | **40** | **3756** |
| WIKI-Small subs. | Softmax | 42.4 | 50.2 | – | – | – | 16500 | 9214 |
| | ST$^*$($k=4$) | 48.7 | 51.5 | 8 | 30 | 4.6 | 36 | 691 |
| | **AST**$^*$($\alpha$=0.35,$\rho$=1.2) | 46.3 | **49.5** | 11 | 73 | 4.1 | **16** | **586** |
| | ST$^*$($k=9$) | 44.1 | 48.3 | 8 | 50 | 8.0 | 27 | 918 |
| | **AST**($\alpha = 0.38, \mu = 0.1$) | 43.7 | **46.9** | 11 | **13** | 44 | 8.4 | **791** |
| | ST($k=13, \mu = 0.1$) | 44.1 | 48.3 | 8 | 13 | 40 | 12.1 | 1104 |
| | **AST**$^*$($\alpha$=0.39,$\rho$=1.2) | 43.6 | **47.5** | 11 | 34 | 11.7 | **12** | **929** |
| | ST($k=67, \mu = 0.01$) | 29.6 | 48.4 | 8 | 21 | 256 | 8.11 | 2291 |
| | ST($k=95$) | 19.7 | 44.1 | 8 | 256 | 5.7 | 30 | 3065 |
| | ST(from AST) | 21.1 | 44.0 | 8 | 65 | 12.5 | 19 | 3296 |
| | **AST**($\alpha$=0.69,$\rho$=1.2) | 37.8 | **42.7** | 13 | 184 | 2.8 | **13** | **1437** |

consisting of more than 36k classes. The full dataset contains roughly 380k features and 800k training samples. Setting the initial depth of AST to small values (2-3) while keeping $\alpha$ relatively high (0.6-0.9) generates extremely big softmaxes in the initial tree, subsequently, causing slow training. Two ways to mitigate this problem: 1) initializing with a bigger initial tree and 2) initializing with smaller $\alpha$ (0.01-0.02) while keeping $\alpha$ in expanding subtrees high (0.7). As a result, as AST expands it covers more and more classes.

Table 2 shows that AST performs better on the test set than most of the baselines. Moreover, ASTs show 6 times faster inference than STs (ASTs contain on average 44 classes in the leaves). Note increasing the number of TAO iterations during leaf expansion or global optimization (or both) may lead to much better results at a cost of training time.

## 6.4 LANGUAGE MODELING

Penn Treebank (PTB) is a popular dataset often used for language modeling. We compare the performance of AST models on this task against Hierarchical Softmax (HSM), STs and linear one-vs-all clasifiers. The details about dataset preprocessing, implementation of the baselines and

the hyperparameter tuning can be found in Appendix A.3.

The perplexity score

$$\text{PPL} = \exp\left(-\frac{1}{N}\sum_{n=1}^{N} \log Pr(y_n|\mathbf{x}_n)\right)$$

can be undefined for models that can output exactly zero probability. This can happen with STs where an instance $\mathbf{x}$ reaches a leaf whose softmax does not specialize in the true class $y$ and thus gets $Pr(y|\mathbf{x}) = 0$. Therefore, in estimating the PPL we only include the instances for which the model outputs nonzero probability. Although a linear classifier provides a positive probability for all the classes, it could not predict correctly 58% of all $K \approx 6k$ classes on both training and test sets, i.e., the outputted score $Pr(y|\mathbf{x})$, though being positive, was not a maximum, not even in the top-10 for many instances. For our AST models it is possible to control the percentage of points for which the model outputs positive probability by tuning the hyperparameter $\alpha$, which appendix A.4 explores in detail.

Table 3 shows the results on PTB. It is clear that our method outperforms other baselines in both top-1 test error and inference time by a considerable margin. The performance of AST can be even further improved by more optimization

Table 2: Results on the text classification dataset WIKI-Small. We report the test error, depth $\Delta$ of the tree, and the average inference time per test sample in milliseconds. For STs we specify the leaf softmax size $k$, and for ASTs we specify the softmax contraction coefficient ($\alpha$) and the tolerance ratio of node expansion ($\rho$).

| Method | $E_{\text{test}}(\%)$ | $\Delta$ | inf.(ms) | Train time |
|---|---|---|---|---|
| RecallTree | 92.64 | 15 | 0.97 | 53m |
| one-vs-all | 85.71 | 0 | 10.70 | > 7d |
| MACH | 84.80 | – | 252.64 | 1445m |
| ST(k = 200) | 84.70 | 8 | 0.18 | ≈1000m |
| $(\pi, \kappa)$-DS | 78.50 | – | 10.33 | – |
| ST(k = 150) | 77.26 | 8 | 0.57 | ≈1000m |
| **AST**($\alpha$=0.6, $\rho$=1.0) | 77.30 | 12 | **0.03** | ≈2000m |
| **AST**($\alpha$=0.60, $\rho$=1.1) | **76.21** | 12 | 0.04 | ≈2000m |

Table 3: Results on the language modeling dataset PTB. We report test error, depth $\Delta$ of the tree, the average inference time per sample in microseconds and the average perplexity (PPL) over the test set instances for which the model outputs nonzero probability. The percentage of such instances is shown in parenthesis. For AST models $\rho = 1.0$.

| Method | $E_{\text{test}}(\%)$ | $\Delta$ | inf.(µs) | PPL | (%nnz) |
|---|---|---|---|---|---|
| HSM | 91.1 | 18 | 421 | 575 | (100%) |
| one-vs-all | 87.5 | 0 | 705 | 220 | (100%) |
| ST(k = 50) | 86.5 | 8 | 58 | 17 | (44%) |
| ST(k = 100) | 86.5 | 7 | 58 | 27 | (51%) |
| ST(k = 400) | 86.4 | 5 | 64 | 71 | (67%) |
| **AST**($\alpha = 0.3$) | 86.4 | 12 | **17** | 10 | (37%) |
| **AST**($\alpha = 0.4$) | **86.1** | 12 | 18 | 13 | (44%) |
| **AST**($\alpha = 0.5$) | 86.2 | 11 | 19 | 24 | (51%) |
| **AST**($\alpha = 0.75$) | 86.3 | 12 | 20 | 7 | (33%) |

iterations.

## 6.5 TREE STRUCTURE AND INTERPRETABILITY

Fig. 3 shows how the number of classes present in the leaves changes with depth. Theoretically, the number of classes in the leaves should only monotonically decrease with depth. Such deviations are due to two reasons: 1) a number of classes in the reduced set of the given depth is lower than the theoretical upper limit; 2) post pruning brings leaves closer to the root.

The built-in tree structure of our model makes it possible to interpret it by visualizing the tree structure and the tree parameters. To show this, we train an AST on a small subset of Amazon Reviews dataset [He and McAuley, 2016] which contains text reviews for the products on the Amazon website. From four high-level product categories (Sports, Toys, Home, Tools) we select 50 subcategories with the highest number of reviews. We select up to 300 reviews from each subcategory and extract tf-idf transformed bag-of-words features. This results in a dataset of size about 60k instances with features of dimensionality 11k and 200 classes. We keep 20% of the dataset as a test set and train a relatively smaller AST on this problem to be able to visualize it in a figure. An initial tree has depth $\Delta_0 = 2$, and $\alpha = 0.25$, and we limit the expansion steps up to 2. The resulting tree has an accuracy of 53%, and depth $\Delta = 6$, and is visualized in fig. 4. At first glance a hierarchical structure is obvious, where we can observe some subtrees specializing in similar groups of classes; for example, decision node 9 specializes mostly in Toy classes. Looking at the decision node weights in any given root-leaf path one can get a local interpretation of why the tree sends a point to that particular leaf. A small and sparse softmax model at the leaf can also be interpreted. Another key observation is that, for the most part, similar classes tend to be grouped within the same leaf,

which is quite remarkable given that the tree is initialized randomly and unaware of any class information.

## 7 CONCLUSION

Softmax Trees are effective for many-class problems by capitalizing on the conditional computation of decision trees and the ability to define local softmax classifiers that handle small subsets of classes, both of which make inference very fast. However, the existing algorithm operates on a fixed, complete tree, which computationally limits the depth of any individual leaf and forces the local softmaxes to be wider than necessary. Our Adaptive Softmax Tree solves this by learning the tree structure, so it can have deeper leaves with thinner softmaxes. It achieves this by interleaving local expansion steps that turn a wide softmax into a softmax subtree with thin softmaxes, with a global TAO optimization of the entire tree. Our experiments convincingly show how this results in improved accuracy, inference time and model size, which makes well worth its longer training time.

**Limitations** Although our algorithm is guaranteed a monotonic decrease of the objective function (at both regular and expansion steps), we lack any other theoretical guarantees of optimality (which are difficult to obtain for alternating optimization methods on nonconvex nondifferentiable problems). Also, while trees making hard decisions result in very fast inference, training them end-to-end with neural networks is not straightforward. One approximate approach is to train a neural network with either a regular flat softmax or a hierarchical softmax and then replacing it with an AST in a teacher-student approach to obtain an overall model with faster inference, as we do in our language modeling experiments.

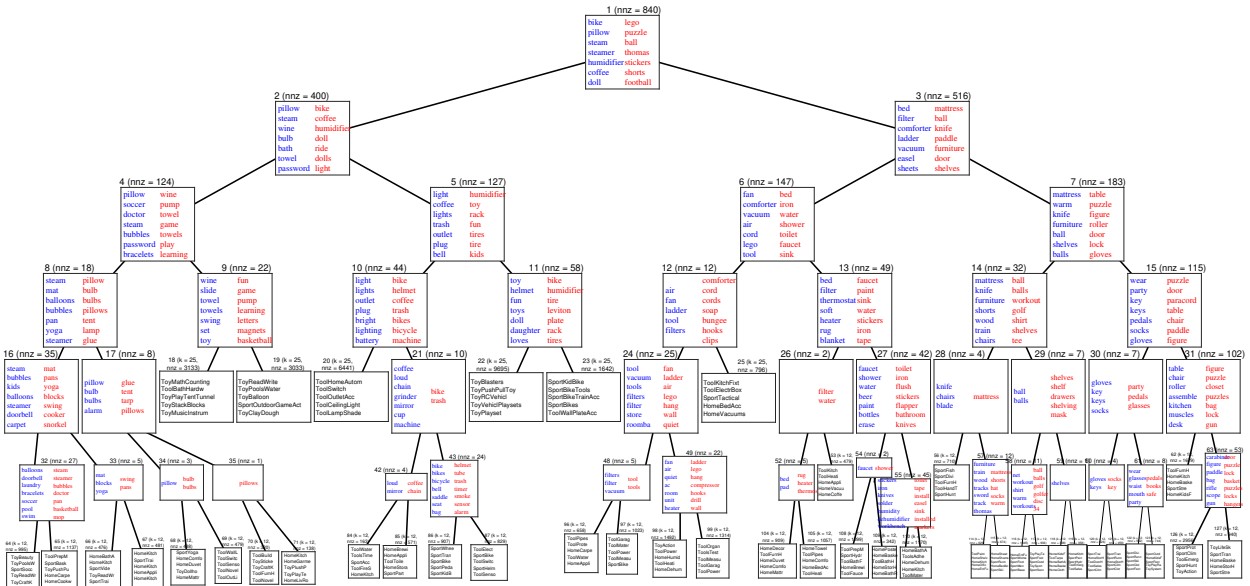

Figure 3: AST for the Wiki-Small subs. dataset. Size of the blue nodes (on the tree) shows the actual number of classes in the leaves after pruning. Green (left column) shows theoretical max. values at each aligned depth.

Figure 4: Visualization of an adaptive softmax tree for a subset of the Amazon Reviews dataset. You may want to zoom in.

**Acknowledgements**

Work partially supported by NSF award IIS–2007147.

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

| Dataset | $N_{\text{train}}$ | $N_{\text{test}}$ | $D$ | $K$ |
|---|---|---|---|---|
| Letter | 16 000 | 4 000 | 16 | 26 |
| ALOI | 97 200 | 10 800 | 128 | 1000 |
| LSHTC1 | 80 552 | 19 873 | 271 022 | 2657 |
| WIKI–Small (subs.) | 20 000 | 10 000 | 54 188 | 200 |
| WIKI–Small | 796 617 | 199 155 | 380 078 | 36 504 |
| PTB | 400 097 | 34 633 | 150 | 5 970 |

Table 4: Datasets used in the experiments: number of train and test instances ($N_{\text{train}}$, $N_{\text{test}}$), number of features $D$, number of classes $K$.

## A APPENDIX

### A.1 DATASETS

To create the subsampled Wiki-Small dataset, we randomly select an equal number of samples from each class to avoid imbalance. This is done with two purposes: 1) a smaller dataset allows training for a much higher number of iterations (to eliminate undertraining); 2) reduces the time of a single experiment which facilitates a more precise hyperparameter search. Further, we remove features that remain constant for all the training and test points. As a result, input features of the subsampled WIKI-Small have $D = 37k$ dimension represented as normalized bag-of-words. For LSHTC1, we eliminate all classes that contain less than 10 samples per class. We used tf-idf feature representations of $D = 271k$ dimension and $K = 2657$ classes. Table 4 summarizes the used dataset statistics. ALOI can be found here https://aloi.science.uva.nl/. Wiki-Small and LSHTC1 are both part of Large Scale Hierarchical Text Classification challenge (LSHTC) [Partalas et al., 2015]. Preprocessed version of PTB dataset from Mikolov et al. [2010] was used. Letter dataset can be found in UCI ML dataset repository [Lichman, 2013].

### A.2 IMPLEMENTATION DETAILS AND HYPERPARAMETERS

Both ST and AST were implemented in Python 3.8.10 and parallelized using Ray 2.2.0 [Moritz et al., 2018]. The $l_1$-regularized logistic regression used the implementation available in `scikit-learn` [Pedregosa et al., 2011]: in the decision nodes we used LIBLINEAR [Fan et al., 2008], and in the leaves we used SAGA [Defazio et al., 2014].

For ST, a search of hyperparameters was performed on a separate holdout set. We found that $\lambda = 0.01$ leads to the best performance for most datasets and $\lambda = 1$ - for WIKI-Small. For smaller datasets we set the number of TAO iterations high (up to 100), we report an average of 5 runs, and set the number of LIBLINEAR and SAGA iterations to 100. For larger experiments, TAO iterations are set to 40 with an average of 3 runs, and the number of LIBLINEAR and SAGA iterations is set to 100 and 50 respectively. Trees were initialized using random initialization as well as k-means initialization described in Zharmagambetov et al. [2021a].

For AST, both leaf and node sparsity parameters were cross-validated separately on a range between 0.01 and 100. It was found that for Letter and subsampled WIKI-Small $\lambda = \mu = 0.01$, and for ALOI and LSHTC1 $\lambda = \mu = 0.1$ performs best. For large datasets, $\lambda = \mu = 1$ produces best results. We initialized the initial tree as well as stumps during expansion using median split. This way nodes have almost the same number of samples and training in parallel becomes faster and generally produces better accuracy. The number of LIBLINEAR and SAGA iterations is similar to one in ST. One way of speeding up the expansion process is to use the weight matrix of the expanding decision node to warm-start optimization in leaves. This way SAGA converges much faster for the same tolerance. The number of TAO iterations during the expansion is set to 10 and to 15 during global optimization, but in many cases, it converges faster.

### A.3 LANGUAGE MODELING EXPERIMENTS

The Penn Treebank contains around 1M tokens and a vocabulary size of 10k words. Similar to Zharmagambetov et al. [2021a], we filter out rare words and obtain word embeddings using pre-trained GloVe [Pennington et al., 2014]. We predict the next word based on the previous 3 words. To form a preprocessed dataset, we simply concatenate word vector representations. As a result preprocessed PTB consists of roughly 400k training samples, 150 features, and 5970 classes.

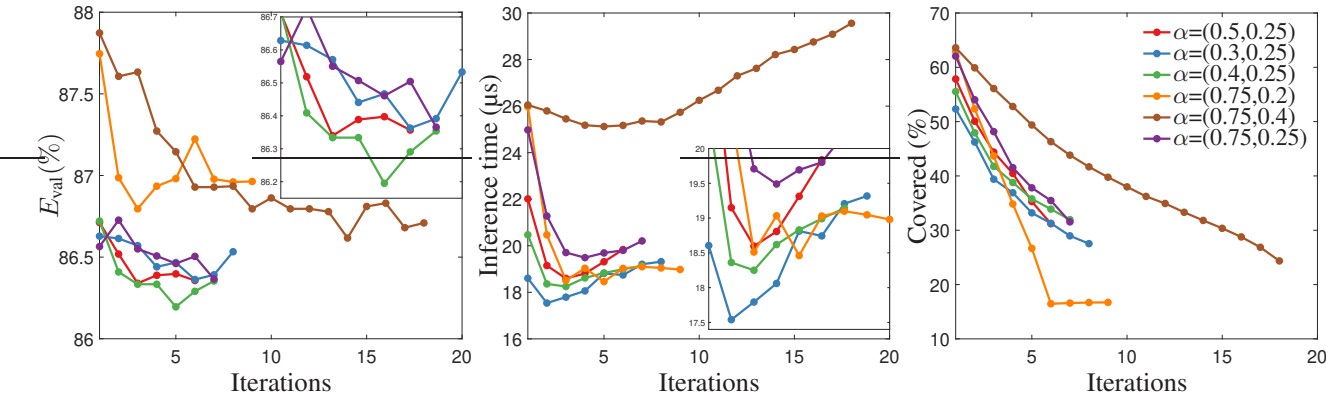

Figure 5: Top-1 error, average inference time and percentage of covered classes for AST of different $\alpha=(\alpha_0, \alpha)$ on PTB dataset.

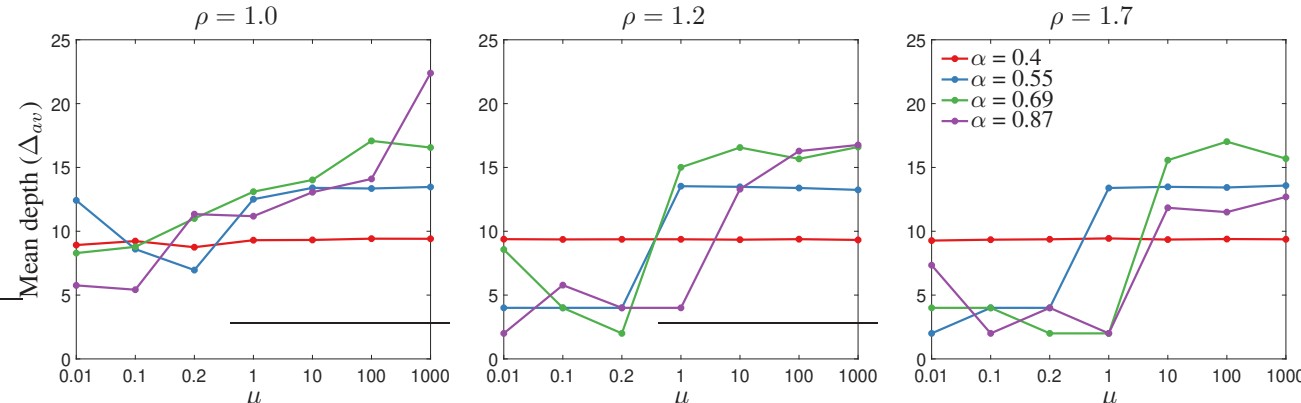

Figure 6: Comparison of average tree depth $\Delta_{av}$ vs softmax regularization parameter $\mu$ for different values of $\rho$ and $\alpha$, $\lambda = 0.01$

For baselines, we used a one-vs-all classifier from `scikit-learn` with $\ell_1$ regularization $\lambda = 1$ and Hierarchical Softmax from Mikolov et al. [2013a] implemented in PyTorch. We further compare AST and ST of different leaf softmax sized ($k$) to show that AST wins not only in terms of top-1 test error but is up to 4 times faster in inference.

## A.4 CONTROLLING LEAF SOFTMAX SIZES FOR LANGUAGE MODELING

Fig. 5 shows dependence between the proportion of covered samples of different AST models. We found experimentally that the best validation performance is achieved when $\lambda$, $\mu$, and $\rho$ are set to 1. Fig. 5 shows that for high values of $\alpha_0$ and $\alpha$ ($\alpha_0 = 0.75$, $\alpha = 0.4$) tree grows extremely deep (high number of expansion steps) while maintaining relatively big softmax in the leaves. Moreover, fig. 5 highlights that as softmax size decreases with tree depth so does the inference time, however, at some point, it starts to go up again. Since the time it takes to propagate a sample to the leaf overtakes the time of matrix multiplication in softmax there is an optimal depth of the tree for which inference is the fastest. On the other hand, for very small $\alpha$ ($\alpha_0 = 0.75$, $\alpha = 0.2$) softmax size decreases much faster with tree depth resulting in a small tree with a very small number of classes in the leaves. Experimentally we found that such trees do not generalize very well and typically have low class coverage. We can specify the number of expansion steps (maximum depth) of the tree to control the minimum coverage and inference time.

We examine the effect of softmax contraction coefficient ($\alpha$), tolerance ratio for node expansion ($\rho$), and leaf sparsity parameter ($\mu$) on the final tree structure. We conducted this set of experiments 5 times on a subsampled WIKI–Small dataset to eliminate the effect of noise and any inconsistencies.

We measure average tree depth $\Delta_{av}$ over the depth of each leaf in the final AST. Fig. 6 shows that $\Delta_{av}$ tends to increase as

we increase $\mu$. More sparse softmax in the leaves means expanded subtree is more likely to perform better on the reduced set. It subsequently leads to more leaves being expanded on the current depth. Maximum depth, on the other hand, does not grow significantly. Fluctuations of average depth as we increase leaf sparsity can be explained by good local optimum for given $\mu$. In general, it was found that as the number of TAO and SAGA (solver for softmax classifier) iterations increases lines become more smooth.