# OpenReview forum: "Adaptive Softmax Trees for Many-Class Classification"
_auai.org/UAI/2024/Conference — UAI 2024 poster_

### Official Review · Reviewer_B7ig · 2024-03-06

**Q2-1 Originality-Novelty:** 3
**Q2-2 Correctness-Technical Quality:** 4
**Q2-5 Clarity Of Writing:** 4

**Q1 Summary And Contributions:**

The article considers a multiclass classification problem with a large number of classes. The article improves upon the earlier softmax trees (ST) by introducing adaptive softmax trees (AST) that enable growing deeper trees with an irregular structure. The process of growing an AST tree starts with a shallow ST tree, and alternates between expansion and reoptimization steps. The experimental results demonstrate that ASTs have a faster inference times and lower test errors than original STs.

**Q2-3 Extent To Which Claims Are Supported By Evidence:**

4: Excellent: all claims are supported by very convincing evidence (in the form of comprehensive experimental evaluation, rigorous mathematical proofs, detailed (pseudo-)code, precise references, well-motivated and realistic assumptions) and the authors deliver what they promise.

**Q2-4 Reproducibility:**

3: Good: key resources (e.g. proofs, code, data) are available and key details (e.g. proofs, experimental setup) are sufficiently well-described for competent researchers to confidently reproduce the main results.

**Q3 Main Strengths:**

The article is clearly written and easy to follow. The proposed algorithm is explained in a sufficient detail. An article gives a very good overview of the related work. An intuition behind the improvement to the earlier method is explained clearly. The experimental results demonstrate a clear improvement, and there are sufficient ablation experiments.

**Q4 Main Weakness:**

As the authors readily admit, the article builds upon an earlier method (softmax tree), and proposes a single improvement to this algorithm, so the contributions of the article are not very wide.

**Q5 Detailed Comments To The Authors:**

Have you tested a simpler version where the width of the softmax is not limited with the parameter alpha, but at each node all of the classes present among the training set points in that node are be considered?  Figure 2 suggests that the actual number of classes in the leafs is often smaller than the upper limit, so it seems that the width of the softmax is limited quite well automatically? If you have tested, what are the tradeoffs? In particular, does setting a hard limit with alpha speed up the training significantly?

As a minor detail, some items in the bibliography are strangely formatted. The proceedings of ICML 2017 and NeurIPS 2017 are referred only with the editors names (e.g., Precup and Teh [2017]). This is confusing. As a related detail, the editors of the conference proeceedings are usually not mentioned in the bibliography. I think that omitting them in all of the items would make the bibliography clearer.

**Q9 Complying With Reviewing Instructions:**

Yes

---

> ### Author Rebuttal · Authors · 2024-04-09
>
> Thanks for providing a very insightful and valuable review. We greatly appreciate your effort!
>
> 1) As the authors readily admit, the article builds upon an earlier method (softmax tree), and proposes a single improvement to this algorithm, so the contributions of the article are not very wide.
>
> **Answer.** The proposed extension fills an important gap in the original ST model. Since the original ST assumes an initial tree structure of a given depth, it cannot explore much deeper tree structures, as their cost grows exponentially with the depth; yet, some regions of input space may required a deeper partitioning. The ST is also sensitive to initial model parameters, necessitating the use of various heuristic initializations (e.g. $k$-means clustering). Our proposed approach (AST) effectively addresses these two shortcomings by expanding the tree where needed and globally reoptimizing the whole model. Each step of our algorithm has a guarantee of monotonically decreasing the objective function value. Furthermore, growth can be undone further during the training process due to pruning during global optimization over tree parameters. In this sense, our proposed approach is a sophisticated search for architectures that are highly irregular and much deeper, while optimizing each fixed-structure model. No other algorithm has done this that we know.
>
> Also, an important goal of the proposed approach is to learn an accurate model that has very fast inference. We prove in section 5.1 Computational Complexity that deeper trees with smaller softmaxes reduce the computational complexity of the tree. We give further empirical evidence by providing both FLOPs and inference time on a single CPU. The training time is longer because of the more exploration and optimization steps, but the significant gain in accuracy and inference time justifies the approach. To train an AST efficiently we use warm starting in decision nodes and in leaf nodes. Furthermore, each expansion step and regular step requires up to 5 - 10 TAO iterations (depending on the dataset), while the original ST is typically trained for much longer. While training time is slower, it is comparable to other methods such as ST and MACH, and much faster compared to softmax or one-vs-all.

---

### Official Review · Reviewer_3GWd · 2024-03-19

**Q2-1 Originality-Novelty:** 3
**Q2-2 Correctness-Technical Quality:** 3
**Q2-5 Clarity Of Writing:** 4

**Q1 Summary And Contributions:**

This work builds further upon Softmax Trees (ST); a recent approach for extreme classification where a decision tree in the form of a complete binary tree is fitted, having linear discriminant functions in the internal nodes and softmax classifiers in the leaf nodes. In this work, a new algorithm -- called Adaptive Softmax Trees (AST) -- is proposed that trains an ST of arbitrary structure (i.e. not restricted to complete binary trees) and therefore improves both in terms of computational complexity and performance during test time.

**Q2-3 Extent To Which Claims Are Supported By Evidence:**

2: Fair: the main claims are somewhat supported by evidence (but the experimental evaluation may be weak, or does not match entirely with the claims, important baselines may be missing, proofs contain important ideas but lack rigor, algorithmic details are only discussed superficially, references are imprecise, assumptions are not sufficiently motivated or explicated, etc.).

**Q2-4 Reproducibility:**

3: Good: key resources (e.g. proofs, code, data) are available and key details (e.g. proofs, experimental setup) are sufficiently well-described for competent researchers to confidently reproduce the main results.

**Q3 Main Strengths:**

The paper is in general well-written and easy to follow, provided that the reader is familiar with the literature on extreme multi-class classification. The proposed algorithm is well-described by means of pseudocode and computational complexity analysis. In addition, the authors provide a concise summary of the limitations, which was appreciated by this reviewer. In addition, the authors illustrate the usefulness of the proposed algorithm, improving over traditional STs, as is clear from the empirical evidence. In addition, the authors provide insights in terms of the learned tree structure after applying AST, which was also appreciated.

**Q4 Main Weakness:**

Although it is clear from the experiments that AST improves over ST, given four different benchmark datasets, the empirical evaluation on the other two datasets (see Table 2) do not show the advantage of AST compared to other existing methods in the literature. For example, the authors do not include a comparison with traditional softmax (in tandem with GPU) and the hierarchical softmax for both datasets. It's also not entirely clear why the four benchmark datasets were used in the second run of experiments.

In addition to my previous comment, the paper fails to show the relevance of the proposed methodology (and the previously-proposed ST). Given the nature of (A)ST, it is clear that the output space of possible probability distributions, given an instance $\mathbf{x}$, is restricted, since a lot of probabilities are assumed to be zero. Zharmagambetov et al. (2021), gives the following argumentation: "We argue that having the classifier output a positive probability (however small) for each class is slow and unnecessary when K is large, because, for any given instance, the majority of classes should indeed have a negligible probability". Although this motivation is valid from a computational complexity point of view, in terms of correctness, I would argue that this observation is restrictive, because the method implies that instances for which the aleatoric uncertainty is high (eg. near-uniform distribution over the classes), cannot be modeled in a correct way (ie. since a lot of classes are assumed to have zero probability) and therefore the predicted probabilities are not reliable.

As stated by the authors, a limitation of the paper is its current lack of theoretical foundations. The following questions arised when reading the paper:

1) What can be said about the solution that is found by TAO? Do you have any finite sample/asymptotic guarantees in terms of the minimiser for (1)?
2) Secondly, given that the cross-entropy loss is optimized, what can be said about the calibration properties of the resulting classifier?

**Q5 Detailed Comments To The Authors:**

Section 2, Related work -- Softmax approximation: the authors claim that inference time in a large softmax is also very large -- for example, in a language model having a large vocabulary. Can the authors showcase this in the empirical results (eg. Table 2)? I wonder how the proposed methodology evaluates against a traditional softmax that is applied on a GPU.

Section 2, Related work -- Decision tree methods: I feel that the explanation of $(\pi,\kappa)$-DS and MACH is currently limited. Given that these two algorithms are used as baseline in Table 2, I believe that some more explanation here could be beneficial.

Section 2, Related work -- Conditional computation: can the authors clarify what is meant with soft trees? Do you mean trees in which nodes correspond to probabilistic classifiers? Why would that result in a lower accuracy? This seems to contradict the empirical results in Table 2 (HSM, right table).

Section 4, page 3: the predictive function $f: \mathbb{R}^{D} \rightarrow [0,1]^{K}$ -> K should be k?

Section 4, page 3: for all other $K-k$ classes $f_{j}$ assigns exactly zero probability-> this relates to my comments above. This sounds very restrictive and needs a clear motivation.

Section 4, page 4: can the authors explain why sparsity can aid in optimizing the model?

Section 4, page 4: Is the separability condition somehow related to (non)decomposable loss function in statistical learning theory?

Section 4, page 4 (right column): "We can ensure a monotonic decrease of the top-level objective (1)" -> I wonder wheter a monotonic decrease is a strong guarantee here, given that the objective in (1) is probably highly non-convex. Can the authors comment on this?

Page 5, figure 1: I'm not used to pseudocode being rendered as a figure. Perhaps the authors may consider the "algorithmicx" package? Secondly, what is $k_{i}$ in the pseudocode of TAO and how is it chosen?

Page 5, last paragraph section 5: Am I correct in my understanding that $\alpha$ and $\rho$ directly influence the computational complexity? Perhaps the author can make this clear here.

Section 6.2, page 6: Is there any reason why the authors don't use the same benchmark datasets of the first set of experiments in the second type of experiments? How were the two datasets chosen here?

Table 1: Is the Softmax deployed on a GPU and are instances processed in batches? What do you mean exactly with Softmax? More details are needed here.

Table 1: Can you explain why there is such a big difference between Delta=11, 12 and 15 in terms of inference time? Moreover, given that the last AST has a slightly bigger depth, why does it improve so much in terms of inference time?

Section 6.3, page 7: Which hierarchy did you use in the experiments for the hierarchical softmax (HSM)?

Section 6.3, paragraph 2, page 7: $Pr(y|\mathbf{x}=0)$ should be $Pr(y\,|\,\mathbf{x})=0$?

Table 2: How was HSM implemented? Does it facilitate the GPU and does it allow batchwise processing? Or does it run over every sample separately? Would be great to see Softmax (batch mode in tandem with GPU) listed here as well, unless the authors believe that this would not be a fair comparison, in which case I would like to hear the motivation.

Table 2: Given the differences in $\alpha$ for AST, I don’t understand why the depth of trees remain fairly constant. I would assume that a lower alpha would result in less shallow trees?

**Q9 Complying With Reviewing Instructions:**

Yes

---

> ### Author Rebuttal · Authors · 2024-04-09
>
> Thanks for providing a very thorough and insightful review. We greatly appreciate your effort!
>
> *Experiments on softmax*
>
> In Table 2, for the WIKI-Small dataset we could not train the traditional flat softmax on both a CPU and a GPU, because it could not fit into GPU memory or training is very slow on a CPU. The dataset is very sparse and high-dimensional ($D = 380k$) and with a large number of classes ($K = 36k$). And GPUs do not support sparse matrix computation. But for the PTB dataset the training is possible, and the results are as follows: 86.9\% test error with 0.467 ms inference time per sample on the CPU. It is orders of magnitude slower at inference than Softmax Trees and with slightly worse accuracy. We do not perform controlled experiments on GPU inference time because this task is more complicated due to a large number of factors: the choice of a GPU model, batch size, memory transfer time between CPU and GPU, etc.
>
> *On assigning zero probabilities*
>
> All models operate based on certain assumptions, and are controlled by hyperparameters. In our case of Adaptive Softmax Trees, the emphasis is on efficient inference, which is achieved by a tree structure that is selectively deep and by having thin softmaxes at the leaves. However, its hyperparameters can be tuned to tailor it to specific datasets by controlling the number of classes per leaf and the depth of the tree---after all, a flat softmax classifier is a particular case of a softmax tree (having depth 0). This flexibility accommodates cases where instances have many nonzero class probabilities while optimizing a tradeoff of accuracy and inference speed. Despite not capturing all probabilities as a flat softmax, our model can still achieve high accuracy while being much faster at inference. Also, the main focus of our paper is on classification tasks involving single ground truth labels, where there is little aleatoric uncertainty.
>
> *More explanation of baselines $(\pi, \kappa)$-DS and MACH*
>
> We will add more details regarding these baselines in the related work in the final version of the paper.

---

### Official Review · Reviewer_GGKH · 2024-03-23

**Q2-1 Originality-Novelty:** 3
**Q2-2 Correctness-Technical Quality:** 2
**Q2-5 Clarity Of Writing:** 1

**Q10 Ethical Concerns:**

Not found

**Q1 Summary And Contributions:**

This paper proposed adaptive softmax trees (ASTs), where it learns jointly the structure and parameters of the tree, by interleaving steps that grow the structure optimally with steps that optimize the parameters of the current structure. This makes it possible to learn ASTs that can grow much deeper but in an irregular way, adapting to the data distribution.

**Q2-3 Extent To Which Claims Are Supported By Evidence:**

2: Fair: the main claims are somewhat supported by evidence (but the experimental evaluation may be weak, or does not match entirely with the claims, important baselines may be missing, proofs contain important ideas but lack rigor, algorithmic details are only discussed superficially, references are imprecise, assumptions are not sufficiently motivated or explicated, etc.).

**Q2-4 Reproducibility:**

3: Good: key resources (e.g. proofs, code, data) are available and key details (e.g. proofs, experimental setup) are sufficiently well-described for competent researchers to confidently reproduce the main results.

**Q3 Main Strengths:**

The problem is interesting and the solution is useful.

**Q4 Main Weakness:**

This paper is not well-written specially section 4. The contribution of the paper is section 4. But this section is only a summary.

**Q5 Detailed Comments To The Authors:**

Extending Section 4 and shorten sections 1 through 3.

In softmax trees, is $\sigma$ a softmax function or sigmoid function?

Why is $f_j$ a function whose range is $K$-dimensional cube? If $\sigma$ is a softmax function, how did you satisfy the constraint that the sum of the probabilities become 1? But, if $\sigma$ is a sigmoid function, no such constraint will be needed.

How did you replace $f_j$ with another shallow tree? How did you train such a tree? Did you use a two-phase training algorithm or a single-phase? How did you specify the depth of each branch of the tree?

**Q9 Complying With Reviewing Instructions:**

Yes

---

> ### Author Rebuttal · Authors · 2024-04-09
>
> Thanks for providing a very insightful and valuable review. We greatly appreciate your effort!
>
> *Extending Section 4 and shorten sections 1 through 3.*
>
> Because our paper relies on recent algorithms/papers which are not widely known (the ST and TAO papers), it is important to review this in detail and provide adequate context and motivation through Sections 1 to 3. Given that the camera ready version allows up to 2 new pages, we will definitely extend Section 4 by adding more explanation and details, and incorporating feedback from the reviews.
>
> *Is $\sigma$ a softmax function or sigmoid function?*
>
> As we specify in the paper, $\sigma(\cdot)$ is a softmax function.
>
> *Range of the function $\mathbf{f}_j$*
>
> In section 4, we define the predictive function of a leaf $j$ as $\mathbf{f}_j(\mathbf{x};\boldsymbol{\theta}_j)\mathpunct{:}\ \mathbb{R}^D \to [0,1]^K$. We will define this more precisely as $\mathbf{f}_j(\mathbf{x}; \boldsymbol{\theta}_j)\mathpunct{:}\ \mathbb{R}^D \to \mathbb{S}^K$, where $\mathbb{S}^K = \\{ \mathbf{x} \in [0,1]^K \mathpunct{:}\ \mathbf{x}^T\mathbf{1} = 1 \\}$. We were not this precise in the paper because we thought it would be clear from the context that the leaves contain softmax classifiers that satisfy this by construction.
>
> *How did you replace $f_j$ with another shallow tree? How did you train such a tree? Did you use a two-phase training algorithm or a single-phase? How did you specify the depth of each branch of the tree?*
>
> Given the current tree structure and parameters, each leaf $j$ receives a subset of training points that we refer to as its reduced set $\mathcal{R}_j$. To replace the leaf's predictive function $\mathbf{f}_j$, or equivalently to expand the tree through the leaf $j$, we train yet another shallow softmax tree $\tau_j$ of depth either 1 or 2 (for most experiments we use depth 1) on this reduced set $\mathcal{R}_j$ using the TAO algorithm. We review the TAO algorithm in Section 3 of the paper and provide its pseudocode in Figure 1. If this trained tree $\tau_j$ improves upon the leaf's predictive function $\mathbf{f}_j$, then we replace the leaf $j$ with this shallow softmax tree $\tau_j$. This corresponds to the expansion step of our algorithm. This is then followed by a regular step where we retrain the whole model using the TAO algorithm.

---

### Official Review · Reviewer_D6qw · 2024-03-24

**Q2-1 Originality-Novelty:** 2
**Q2-2 Correctness-Technical Quality:** 3
**Q2-5 Clarity Of Writing:** 3

**Q1 Summary And Contributions:**

This paper proposes to build the softmax trees (ST) adaptively, which can train an ST of arbitrary structure, adapting to the data distribution.

**Q2-3 Extent To Which Claims Are Supported By Evidence:**

3: Good: the main claims are supported by convincing evidence (in the form of adequate experimental evaluation, proofs, (pseudo-)code, references, assumptions).

**Q2-4 Reproducibility:**

3: Good: key resources (e.g. proofs, code, data) are available and key details (e.g. proofs, experimental setup) are sufficiently well-described for competent researchers to confidently reproduce the main results.

**Q3 Main Strengths:**

1. The writing is easy to follow.

2. It seems that the empirical results show the effectiveness of the adaptive STs.

**Q4 Main Weakness:**

1. Limited novelty. The proposed method is a margin improvement largely based on previous softmax trees. The new contents start from Section 5 on page 4. Also, the design is empirical without much theoretical contribution.

2. I'm curious about the practical applicability of softmax trees in large-scale training scenarios. In modern DNNs, the softmax layer typically comprises only a small fraction of the overall parameters. Could the efficiency introduced by softmax trees offset any potential decrease in accuracy?

**Q5 Detailed Comments To The Authors:**

See above.

**Q9 Complying With Reviewing Instructions:**

Yes

---

> ### Author Rebuttal · Authors · 2024-04-09
>
> Thanks for providing a very insightful and valuable review. We greatly appreciate your effort!
>
> 1) Limited novelty. The proposed method is a margin improvement largely based on previous softmax trees. The new contents start from Section 5 on page 4. Also, the design is empirical without much theoretical contribution.
>
> **Answer.** The proposed extension fills an important gap in the original ST model, as detailed in section 3. Since the original ST assumes an initial tree structure of a given depth, it cannot explore deeper tree structures; and a complete tree cannot be made very deep due to its computational cost. ST is also sensitive to initial model parameters, necessitating the use of various heuristic-based initializations. Our proposed approach (AST) effectively addresses these two shortcomings by cleverly expanding the tree where needed and globally reoptimizing the whole model. Each step of our algorithm has a guarantee of monotonically decreasing the objective function value. Furthermore, parts of the tree that were grown at some point can be undone later on during the training process, due to (implicit) pruning that can occur during the global optimization over the tree parameters (which have an $\ell_1$ penalty). In this sense, our proposed approach is a sophisticated search for architectures that are irregular and much deeper, while optimizing globally the model for a fixed architecture. No other algorithm has done this that we know.
>
> The goal of the proposed approach is to learn an accurate model with very fast inference. We prove in section 5.1 Computational Complexity that deeper trees with smaller softmaxes reduce the computational complexity of the tree at inference time. We give empirical evidence for this by providing both FLOPs and inference time on a single CPU. The training time is longer because of the more exploration and optimization steps but 1) the significant gain in accuracy and inference time justifies the approach, and 2) we would simply not be able to use the original TAO algorithm on deep complete trees. To train an AST efficiently we use warm starting in the decision nodes and leaf nodes. Furthermore, each expansion step and regular step requires up to 5 - 10 TAO iterations (depending on the dataset), while the original ST is typically trained for much longer. While the training time is slower, it is comparable to other methods such as ST and MACH, and much faster compared to softmax or one-vs-all.

---

### Official Review · Reviewer_UJex · 2024-03-31

**Q2-1 Originality-Novelty:** 3
**Q2-2 Correctness-Technical Quality:** 2
**Q2-5 Clarity Of Writing:** 4

**Q10 Ethical Concerns:**

NA.

**Q1 Summary And Contributions:**

The paper considers the problem of growing a tree that may be optimized with the relatively new Tree Alternating Optimization (TAO) algorithm. This is a novel contribution since TAO works over a fixed complete binary tree and can only extract a subtree out of it. The technique proposed here manages to evolve the structure, while optimizing it, by alternating between substituting a leaf by a subtree optimized with TAO (where certain criteria are met), and then globally optimizing the augmented the tree using TAO.

**Q2-3 Extent To Which Claims Are Supported By Evidence:**

3: Good: the main claims are supported by convincing evidence (in the form of adequate experimental evaluation, proofs, (pseudo-)code, references, assumptions).

**Q2-4 Reproducibility:**

2: Fair: key resources (e.g. proofs, code, data) are unavailable but key details (e.g. proof sketches, experimental setup) are sufficiently well-described for an expert to confidently reproduce the main results.

**Q3 Main Strengths:**

The paper addresses an important problem in DT optimization - that of first-order optimization of a DT (not new) *while* growing the DT (new). The problem is well-motivated and the paper is clearly written in general.

**Q4 Main Weakness:**

I am not sure if the two alternating steps of global optimization and local expansion actually do not end up in conflict with each other.

**Q5 Detailed Comments To The Authors:**

I found the problem clearly motivated and paper well-written.

I have just one major reservation with the proposed technique.
Broadly these are the steps:
* Substitute a leaf $j$ with a subtree $\hat{\tau}_j$, *if* the latter decreases the loss $L$ by a margin (controlled by $\rho$). $\hat{\tau}_j$ is itself optimized using TAO, before the losses are compared.
* After making such a substitution, globally optimize the augmented tree using TAO.

It seems like $ \hat{\tau}_j $ gets two sets of parameters: one, when it is individually optimized (I'll refer to this version of the subtree as
$\hat{\tau}_j^{pre} $), and two, when the global optimization occurs (I'll refer to this version of the subtree as $\hat{\tau}_j^{post} $).

My question is: while the suitability of $ \hat{\tau}_j $ is gauged based on $\hat{\tau}_j^{pre} $, what really ends up in the overall tree is $\hat{\tau}_j^{post} $. Given this I am not sure what role the assessment based on $\hat{\tau}_j^{pre} $ exactly serves. To posit an extreme for making a point: if I tried replacing each of the leaves with a randomly initialized subtree structure, performed a global TAO, and *then* decided whether we keep the subtree or not, would I end up with better, or on par, or worse results than the proposed technique?

Can the authors clarify this?

Other:
* Section 2, *Conditional computation*, wrt "Although several works Shazeer..", the citation style should be "..[Shazeer...".
* Section 2, *Growing neural...*, last line "the trees are soft...and very small..". I don't understand why the small trees are a problem.
* Page 3, col 2, "..the inference time is $O(D(\Delta + K))$...". Note that $D$ hasn't been defined yet.
* Above Eqn (1), "...the goal of TAO is to optimize". Always good to be specific where we can, "optimize" --> "minimize".
* In Table 1, what does $\bar{k}=1.01$ signify? How many classes are these softmaxes handling?
* In Section 3.1, it is mentioned "...the softmax size of ST* ($k=13$) ....". Where is this in Table 1?
* In Section 6.3, it is mentioned "...we only include instances .....model outputs nonzero prob...". What is the proportion of these examples?

**Q9 Complying With Reviewing Instructions:**

Yes

---

> ### Author Rebuttal · Authors · 2024-04-09
>
> * Q4. I am not sure if the two alternating steps of global optimization and local expansion actually do not end up in conflict with each other.}
>
> * Q5 If I tried replacing each of the leaves with a randomly initialized subtree structure, performed a global TAO, and then decided whether we keep the subtree or not, would I end up with better, or on par, or worse results than the proposed technique?}
>
> **Answer to both**
>
> Thanks for this question, which helps clarify how our algorithm works. A fundamental idea in our approach is 1) to be able to do a good optimization over a fixed-structure softmax tree (with a fixed set of parameters), while 2) exploring the vast space of possible tree structures in a local way that improves the current tree. Point 1) is readily achieved using the original TAO algorithm, which is guaranteed to decrease the objective function monotonically at each iteration. Point 2) needs a controlled expansion move: we must be able to compare the objective function before and after to decide whether or not we should pursue a new architecture, and this is possible by expanding a leaf subtree and optimizing it separately. The reason is the separability condition that trees satisfy: the objective function separates additively over the leaf subtrees, because each leaf subtree operates only on its own parameters, its own region of the input space and its own reduced set (training instances reaching the leaf). Thus, the contribution to the overall (tree-wide) objective function of optimizing over an expanded leaf subtree is a separable term. Comparing the loss on the leaf reduced set before (softmax) and after (softmax subtree optimized only on that reduced set), together with the penalty on the number of parameters, gives the exact improvement of the overall objective function. Essentially, we are making an apples-to-apples comparison of two models (original and expanded) on the same region and training subset, which translates into an apples-to-apples comparison of the overall objective function given all its other parameters are fixed. Importantly, this makes it possible to decide locally whether to accept the expansion or not. And this can be done in parallel, independently, for all leaves. The subsequent, global optimization with TAO of the expanded tree may, of course, undo some of the expansions, as well as update all the parameters and reduced sets.
>
> Our approach is, however, not the only way to search for tree structures. Indeed, one could do as you suggest: expand all leaves using random subtrees and then optimize the tree globally. This is more costly in that it has to optimize globally a bigger tree (with our approach, we observe many leaves do not accept the expansion move). Also, it does not allow us to make a local apples-to-apples comparison at each leaf, because after the global optimization the reduced set at each leaf may be different from before. We did not explore this approach.
>
> Empirically, we find our approach does a good exploration of the structure space, leading to trees that are more accurate than a complete softmax tree and have a highly irregular (but locally optimal) structure, as demonstrated in Figure 2. The local expansion moves are fast thanks to using the existing weight matrix to warm-start the optimization in the leaves. As a result we keep the number of TAO iterations low in both expansion and global optimization. Also, the hyperparameter $\rho$ helps to control overfitting as the tree grows, which in the case of a random expansion of all the leaves is more challenging.
>
> Other:
>
> * -In Table 1, what does $\bar{k}=1.01$ signify? How many classes are these softmaxes handling?
>
> **Answer.** $\bar{k}$ shows the average leaf softmax size in the tree. In this case, most of the softmaxes contain only 1 class, indicating that tree grows till most leaves are pure (contain one class).
>
> * -Section 2, Growing neural..., last line "the trees are soft...and very small..". I don't understand why the small trees are a problem.
>
> **Answer.** We meant small trees are a problem for achieving fast inference. If a neural tree is small, it will have a relatively large neural net in each node for it to achieve high accuracy, so even if we follow a single path (as an fast approximation) it will still be computationally costly. A deeper tree, having many, deep, lighter leaves, is faster at inference, and can have high accuracy.
>
> * -In Section 3.1, it is mentioned "...the softmax size of ST* ($k=13$) ....". Where is this in Table 1?
>
> **Answer.** $ST(k=13)$ is a mistake, we will fix it in the final version.
>
> * -In Section 6.3, it is mentioned "...we only include instances .....model outputs nonzero prob...". What is the proportion of these examples?
>
> **Answer.** We show the percentage of the covered samples in Appendix A.4 Figure 3.
>
> Thanks for your other comments, which we can address easily in the final version of the paper.

---

### Meta-Review · Area_Chair_MX3j · 2024-04-22

I quite enjoyed the premise of the paper. The idea is to take a very old idea, the hierarchical softmax, and redo it. I think, in general, using a tree to express a softmax reduces the computation from O(N) to O(log(N)) for N elements in the vocabulary. The idea here is to use an algorithm to find something in between, which speeds up the final bottleneck of a neural network. The reviewers agreed the paper was technically sound so I believe it should be accepted.